# Adipose Tissue and Adipose-Tissue-Derived Cell Therapies for the Treatment of the Face and Hands of Patients Suffering from Systemic Sclerosis

**DOI:** 10.3390/biomedicines11020348

**Published:** 2023-01-26

**Authors:** Anouck Coulange Zavarro, Mélanie Velier, Robin Arcani, Maxime Abellan Lopez, Stéphanie Simoncini, Audrey Benyamine, Quentin Gomes De Pinho, Raphael Coatmeur, Jiucun Wang, Jingjing Xia, Ludovica Barone, Dominique Casanova, Françoise Dignat-George, Florence Sabatier, Brigitte Granel, Jérémy Magalon, Aurélie Daumas

**Affiliations:** 1INSERM, INRA, C2VN, Aix-Marseille University, 13005 Marseille, France; 2Cell Therapy Department, Hôpital de la Conception, AP-HM, INSERM CIC BT 1409, 13005 Marseille, France; 3Internal Medicine, Geriatric and Therapeutic Department, Hôpital de la Timone AP-HM, 13005 Marseille, France; 4Plastic Surgery Department, Hôpital de la Conception, AP-HM, 13005 Marseille, France; 5Internal Medicine Department, Hôpital Nord AP-HM, 13015 Marseille, France; 6Human Phenome Institute, Collaborative Innovation Center for Genetics and Development, Fudan University, Shanghai 200438, China; 7Greater Bay Area Institute of Precision Medicine, School of Life Sciences, Fudan University, Guangzhou 511462, China; 8Department of Biotechnology and Life Sciences, University of Insubria, 21100 Varese, Italy

**Keywords:** systemic sclerosis, adipose tissue, adipose-tissue-derived cell therapy, fat grafting, stromal vascular fraction, adipose-derived stromal/stem cell

## Abstract

Adipose tissue is recognized as a valuable source of cells with angiogenic, immunomodulatory, reparative and antifibrotic properties and emerged as a therapeutic alternative for the regeneration and repair of damaged tissues. The use of adipose-tissue-based therapy is expanding in autoimmune diseases, particularly in Systemic Sclerosis (SSc), a disease in which hands and face are severely affected, leading to disability and a decrease in quality of life. Combining the advantage of an abundant supply of fat tissue and a high abundance of stem/stromal cells, fat grafting and adipose tissue-derived cell-based therapies are attractive therapeutic options in SSc. This review aims to synthesize the evidence to determine the effects of the use of these biological products for face and hands treatment in the context of SSc. This highlights several points: the need to use relevant effectiveness criteria taking into account the clinical heterogeneity of SSc in order to facilitate assessment and comparison of innovative therapies; second, it reveals some impacts of the disease on fat-grafting success; third, an important heterogeneity was noticed regarding the manufacturing of the adipose-derived products and lastly, it shows a lack of robust evidence from controlled trials comparing adipose-derived products with standard care.

## 1. Introduction

Systemic Sclerosis (SSc, scleroderma) is a rare connective tissue disease characterized by microvascular injury, an inflammatory and dysimmune state and a progressive fibrotic process involving the skin and deep organs [1]. Although the exact cause of SSc remains undetermined, genetic susceptibility and environmental factors trigger disease onset with (1) early endothelial damage and small vessel vasculopathy, (2) SSc-related autoantibody production and (3) fibroblast activation leading to collagen and extracellular matrix protein production, with progressive fibrosis of the skin and internal organs. New classification criteria for SSc were established by EULAR and ACR [2]. A score of 9 or more permits classifying patients as having SSc. Two major disease subtypes based on the extent of skin involvement are described: the diffuse cutaneous SSc (dcSSc), with skin thickening extending to the trunk and the proximal part of the limbs, and the limited cutaneous SSc (lcSSc), where skin tightening is limited to the face, hands and feet [1]. The burden of non-lethal complications associated with SSc is substantial and represents a real challenge for physicians treating SSc patients. Main patients’ concerns rely on their hands and face, with changes in their physical appearance, as well as disabilities affecting daily activities and frequently leading to psychological and social distress. Indeed, these manifestations are very difficult to hide and lead to a worsening of quality of life [3,4,5]. Facial symptoms encompass aesthetic disfigurement, limited expression with mask-like stiffness of the face, a decrease in mouth opening that interferes considerably with life’s basic functions (such as eating, speaking, oral hygiene and professional dental care), and skin changes (telangiectasias, pigmentation disorders, skin atrophy and perioral rhagades). Due to Raynaud's phenomenon, digital ulcers, skin sclerosis, inflammatory arthritis, contractures and hand involvement can induce intense pain and difficulty in performing daily living activities (such as dressing, eating, body hygiene and applying make-up).

To date, no treatment can significantly reverse skin fibrosis or improve hand and face motion and facial appearance. This situation is a real challenge for physicians and innovative therapy is a crucial hope for patients who often feel that these aspects of their disease are neglected. 

The use of adipose tissue as a filling product in plastic and aesthetic surgery has emerged in the 19th century. Autologous use of adipose tissue constitutes an ideal biocompatible filler. Trophic properties have been attributed to the presence in adipose tissue of Adipose Tissue-derived Stromal Vascular Fraction (AD-SVF), including mesenchymal-like stem/stromal cells. Thus adipose tissue is a valuable source of cells with reparative, angiogenic, antifibrotic, and immunomodulatory properties [6,7,8,9]. With the advantage of an abundant supply of fat tissue, ease of harvest by liposuction, and high abundance of stem/stromal cells (compared to bone marrow), fat grafting and adipose-tissue-derived cell-based therapies are becoming attractive therapeutic options in SSc, especially for fibrosis and ischemic manifestations [10].

After a summary of the functional properties of Adipose-derived Stromal/stem Cells (ASCs) and AD-SVF, we reviewed the current literature focusing on the clinical use of autologous fat-grafting and adipose tissue cell-based therapies for face and hands treatment of patients suffering from SSc.

## 2. Biological Properties of ASCs and AD-SVF

AD-SVF represents a heterogeneous cell product that is considerably enriched in adipose-derived stromal/stem cells (ASCs) but also contains endothelial progenitor cells, pericytes, leukocytes and macrophages [11]. 

The efficacy of ASCs is mainly based on paracrine mechanisms through the release of numerous bioactive molecules (cytokines, growth factors) as well as the production of extracellular vesicles rich in mRNA, miRNA and proteins [12,13]. ASCs and/or their conditioned media limit the apoptosis of different cell types, such as cardiomyocytes, muscle cells, renal epithelial cells or endothelial cells [14,15,16,17]. Several factors mediate their antiapoptotic potential, including Vascular Endothelial Growth Factor (VEGF), Transforming Growth Factor β1 (TGF β1), Hepatocyte Growth Factor (HGF), Fibroblast Growth Factor (FGF) or even Insulin-like Growth Factor-1 (IGF-1) [16,17]. 

The proangiogenic effect of ASCs is based on both a powerful paracrine effect [17,18] and a capacity to differentiate into endothelial cells [19]. Indeed, ASCs secrete a large panel of proangiogenic molecules (VEGF, FGF-2 and HGF) that will intervene from the recruitment of endothelial cells to ischemic sites to the stabilization of the neovessels [20]. Furthermore, the crucial role of ASCs in promoting angiogenesis has been demonstrated in several animal models of limb and brain ischemia or myocardial infarction with encouraging results [21,22,23,24].The demonstration of the antifibrotic potential of ASCs is based almost exclusively on in vivo studies. The antifibrotic effect of ASCs was assessed in two preclinical models of SSc (bleomycin-induced model and scleroderma graft-versus-host disease model) with positive results [25]. These studies showed that intravenous injections of ASCs reduce inflammatory infiltrates as well as fibrosis markers such as collagen content. Maria et al. also reported the efficacy of ASCs in a hypochlorous acid-induced mouse preclinical model of SSc [26,27]. In this work, the administration of ASCs reduces fibrosis damage from a histological point of view. The authors show a disappearance of the ASCs seven days after the injection, whereas the peak of action was observed at 3 weeks post-treatment, suggesting that the effectiveness of the ASCs does not rely solely on the mechanisms of cell migration and differentiation. In total, ASCs seem to be able to act at each stage of the development of fibrosis [28]: they limit the early inflammation induced by hypochlorous acid, then reduce extracellular matrix deposits during the establishment of fibrosis and finally promote matrix remodelling by matrix metalloproteinases during the resolution of lesions. These results support the plasticity of ASCs and their ability to respond positively to different pathogenic stimuli.The immunomodulatory effect of ASCs has been demonstrated with the in vitro test called “mixed lymphocyte reaction”, showing an inhibitory effect on the proliferation of activated immune cells in co-culture models with ASCs [29]. The action of ASCs on the immune system is twofold, based on both a direct interaction with immune cells and a paracrine effect. ASCs interact with all the players of the innate (NK cells, macrophages, dendritic cells) and adaptive immune systems (T and B lymphocytes), resulting in a global modulation of immunity [30].

Although the mechanisms underlying the therapeutic efficacy of AD-SVF are not fully elucidated, it appears that the mechanisms of action are predominantly paracrine. Angiogenic properties have naturally been attributed to AD-SVF as it gathers all cell types from the microvasculature of adipose tissue. Indeed, AD-SVF cells spontaneously self-assemble into a hierarchical and robust vascular structure [31]. In addition, the seeding of AD-SVF cells on a Matrigel^®^ matrix resulted in the formation of pseudo-tubes [32]. The depletion of endothelial cells led to a significantly reduced vascularization, comforting the major role of endothelial populations in the proangiogenic activity of AD-SVF. In addition, AD-SVF cells also have immunomodulatory properties. AD-SVF cells can limit tissue damage through the secretion of antiapoptotic and anti-inflammatory factors and modulate immune responses by different mechanisms [33]. Few studies demonstrated the anti-fibrotic potential of AD-SVF, mainly attributed to the ASCs subset [34,35]. The evaluation of several cell-engineered products (microfat, AD-SVF and platelet concentrates) in a preclinical model of skin fibrosis induced by bleomycin confirmed the capacity of AD-SVF to limit the development of fibrosis [36]. Moreover, a phase I clinical trial of AD-SVF injection into scarred vocal cords reported improved clinical scores, suggesting that the AD-SVF displays antifibrotic properties [37]. 

## 3. Clinical Use of Autologous Fat-Grafting and Adipose Tissue Cell-Based Therapies

To identify the existing clinical research on autologous fat-grafting and adipose tissue cell-based therapies, we performed an open literature review. Two independent reviewers (A.D. and M.V.) utilized PubMed (2000 to December 2022) to critically appraise the use of adipose-tissue-derived products in SSc. The search strategy was based on the following keywords and only article published in English with abstract were selected: (fat transfer or fat transplantation or fat grafting or autologous fat-grafting or autologous microfat graft or lipofilling or lipotransfer or stromal vascular fraction or adipose-derived stromal vascular fraction or adipose-derived stromal/stem cell or adipose-derived stem cell or adipose-derived stromal/stem cell or adipose-tissue-derived cell therapy or adipose-tissue-derived cell therapy or adipose-derived regenerative Cell Transplantation) and (Systemic sclerosis or Scleroderma or Raynaud phenomenon).

In total, 129 studies were identified using our search strategy. Studies were eligible if they involved human subjects, regardless of the level of evidence. Studies about the treatment of morphea or localized scleroderma were excluded. We also excluded case reports, reviews, meta-analyses, expert opinions and editorials. After screening the title and abstract, only 21 studies were retained, including two long-term cohort follow-ups. A comprehensive study attrition diagram is provided in Figure 1.

## 4. Main Outcome Measures for the Face and Hands Used in Systemic Sclerosis Clinical Trials

Outcome measures are important in the assessment of SSc patients in practice and research, such as clinical trials, and many patient-reported outcomes can be useful for monitoring SSc patients.

### 4.1. Main Outcome Measures to Assess Facial Involvement

Involvement of the face is a source of functional, aesthetic, and social discomfort, causing significant psychological stress for patients [4,38]. Numerous scores and tests are used in clinical trials to assess the therapeutic outcome:(1)The mouth-related disability can be assessed using the MHISS (Mouth Handicap in Systemic Sclerosis) scale, which is the first mouth-specific disability outcome measure designed for SSc patients [4]. This scale evaluates three domains: reduced mouth opening, sicca syndrome, and aesthetic concerns, with 12 items (each scoring 0 to 4) and a total score ranging from 0 (no mouth disability) to 48 (severe mouth disability). The MHISS score was used to explain that facial disability contributes up to 36% of the variance of the global HAQ (Health Assessment Questionnaire) score, highlighting its burden on the patient’s life and the need for therapeutics [4].(2)Skin involvement is usually assessed using the modified Rodnan skin score (mRSS) [39]. This semi-quantitative score rates the severity of skin sclerosis from 0 (normal skin texture and thickness) to 3 (severe thickness with the inability to pinch the skin into a fold). Measurements can be made at several fixed facial and perioral sites making sure the facial muscles are relaxed. To assess skin hardness, the durometer is an objective, effective, and reliable method. Durometer-measured skin hardness correlates well with mRSS and ultrasound-measured skin thickness [40]. The cutometer is also a non-invasive biometrological method for monitoring skin stiffening [41].(3)Xerostomia can be easily measured using the sugar test (the time it takes for a sugar cube to melt on the tongue without crunching it) and the Xerostomia Inventory Index [42]. The Xerostomia Inventory index is an 11-item summated rating scale that results in a single continuous scale score representing the severity of chronic xerostomia with five levels of answers with responses from “Never” to “Always” (total score range: 11 to 55 extremely dry mouth) [42].(4)Mouth opening is assessed in centimeters by measuring the distance between the tips of the upper and lower incisors with calipers.(5)Standard or 3D photographs can also be used for assessing volumizing and aesthetic effects.

### 4.2. Main Outcome Measures for Hand Assessment

The origin of hand disability is multifactorial and includes vascular alteration with a chronic ischemic process, skin sclerosis, tendon retraction, joint involvement and calcinosis. Each of these lesions causes pain, functional impairment, aesthetic issues, and psychological distress [5]. The main clinical measures for evaluating hand involvement include: (1)The Cochin Hand Function Scale (CHFS), a functional disability questionnaire about daily activities, has been validated for SSc patients. It includes 18 items with five subscales: kitchen, dressing, hygiene, office, and other. Each question is on a scale of 0 to 5. The total score is obtained by adding the scores of all the items (range 0 to 90). It was shown that hand-functional disability was the major component of global disability, contributing to 75% of global disability (HAQ) in SSc patients [5].(2)The mRSS applied to hands, which evaluates skin thickening on the dorsal aspect of the hand and the first and second phalanges of the most affected finger (score 0–18) [39],(3)A visual analog scale (VAS) of hand pain.(4)For global hand mobility: (a) the Kapandji score assessing opposition of the thumb [43]; (b) the Hand Mobility in Scleroderma index (HAMIS), a nine-item performance-based measure of impairment using different grips and movements to assess finger and wrist mobility or the modified HAMIS with only four items [44]; (c) the lateral range of motion of the fingers measured by the distance between the thumb and index finger (1st corner) and the distances between the four fingers (2nd, 3rd, and 4th corners) upon maximal stretch and (d) finger flexion by measuring the finger-to-palm distance in maximal active flexion.(5)The grip and pinch strength using a Jamar dynamometer under standard conditions.(6)Vascular components are assessed by Raynaud’s Condition Score (RCS) and Digital Ulcer (DUs) counts and aspects. The RCS is a daily self-assessment of Raynaud’s phenomenon activity, using a scale of 0 to 10 that incorporates the frequency, duration, severity, and impact of Raynaud’s phenomenon attacks [45]. Outcomes with DUs are not standardized. DU measurements can include the number of DUs, size, pain scale, healing or partial healing of DU (all or a cardinal ulcer). Nailfold capillaroscopy can also be considered as a fairly sensitive and highly specific test for detecting and monitoring scleroderma-spectrum disorders.(7)The Scleroderma Health Assessment Questionnaire (SHAQ) consists of the HAQ (a self-reported questionnaire in eight domains) and a continuous VAS regarding pain, global patient assessment, vascular, digital ulcers, lung involvement, and gastrointestinal involvement [46].

## 5. Particular Aspects of the Use of Autologous Fat-Grafting and Adipose-Tissue-Derived Cell Therapies for Patients Suffering from Systemic Sclerosis 

When faced with a patient suffering from SSc, it is crucial to carefully assess the possible fat harvesting areas and select the technique allowing the best yield based on the experience of the surgeons to obtain enough adipose tissue for clinical use. SSc patients frequently have a low body mass index due to the disease itself with frequent digestive tract involvement. Multiple fat harvesting sites can be required depending on the amount of adipose tissue desired. For example, in our experience [47], a minimum starting lipoaspirate volume of 100 mL was required to obtain 5 mL of AD-SVF using the automated processing CelutionR800/CRS system (Cytori Therapeutics, San Diego, CA, USA). Tsekouras et al. [48] showed some differences in fat composition: outer thigh adipose tissue has significantly higher AD-SVF cell counts and thigh adipose tissue has a significantly higher number of ASCs in comparison to the abdominal, waist, and inner knee lipoaspirates. In our experience, preferred harvesting areas include the inner side of the knees, abdomen, external thigh, internal thigh, and back [47,49]. In cases of previous adipose tissue harvesting or cryolipolysis, a nontreated area is preferred to avoid any fibrosis tissue. 

The severity of skin fibrosis can also hamper fat tissue reinjection, particularly on the face and in the fingers. Indeed, the macrofat harvested according to the Coleman lipostructure^®^ technique with a specific cannula (3 mm, 2 holes over 1 mm, blunt tip) followed by centrifugation (3000 rpm for 3 min) [6] does not enable direct reinjection at subdermal level or in layers that are poorly extensible or fibrotic. An alternative strategy for lipotransfer in the face is the administration of microfat grafting, developed in our centre [50]. To harvest microfat, the cannula used for macrofat is replaced by a multi-perforated cannula (st'RIM cannula, 2 mm, eight holes of less than 1 mm, blunt tip). Therefore, the fat tissue obtained is more fluid allowing the reinjection of microlobules of adipose tissue (the size of the fat lobules varied from 500 µm to 1000 µm versus a size exceeding 1000 µm with Coleman’s technique) using a 21-gauge cannula. With this technique, fat injection resulted in a decrease in skin fibrosis and improved skin vascularization in a bleomycin-induced mouse model of SSc skin fibrosis [50]. Moreover, in vitro viability and migration of isolated ASCs obtained from micro-harvested lipoaspirates were significantly higher than with conventional fat harvesting and the adherence rate of ASCs isolated using the microfat harvesting technique onto matrices was significantly higher [51].

Regarding the fingers, we were concerned with the risk of inducing ischemia through the reinjection of a large volume of fat tissue, thus causing high pressure. To limit this risk, Del Papa et al. [52,53] made a small skin incision at the base of the fingers and Bank et al. [54] injected it in the dorsum of the hand, in the snuffbox, in dorsal and volar webspaces, and along the superficial palmar arch. The risk of ischemia justified our policy of using the AD-SVF for hand therapy instead of fat tissue, as this cell population can easily be extracted from adipose tissue and re-suspended in a final fluid solution allowing subcutaneous injection via a very small caliber cannula without an incision [47]. 

Concerning adipose-tissue-processing techniques, the most commonly reported clinical use in SSc patients are sedimentation (decantation), centrifugation, and filtration (Figure 2).
Lipoaspirate processing by centrifugation necessitated placing the fat in a centrifuge and spun at a specified speed for 3 min. This allows the separation of lipoaspirate into three layers: the superior oily layer, the middle fat layer and the inferior aqueous layer consisting of blood and infiltration liquid. In addition to the three layers, a pellet of cells and debris can be seen at the bottom of the syringes. Only the middle layer is used for grafting.Lipoaspirate processing by sedimentation (gravity separation) simply necessitates leaving the adipose decantate under gravity for approximately 10 to 15 min. This allows to naturally separate lipoaspirate into three layers, as described above. Similarly, the middle layer of the refined fat tissue is reserved for fat injection.Lipoaspirate processing by filtration allows for the removal of fluids, oil and debris faster than decantation and without using centrifugation.

To date, there is no consensus on the best way of processing the fat to ensure maximum retention and viability of the graft. However, Zhu et al. [55] demonstrated that washing with filtration using the Puregraft^TM^ system produces a fat graft with higher tissue viability and lower presence of contaminants compared with grafts prepared by sedimentation or Coleman centrifugation. Additionally, Gerth et al. [56] showed that autologous fat processed using closed-membrane filtration had a significantly higher long-term retention rate than centrifuge-processed fat injected by the same surgeons. Considering the importance of regenerative cells for graft survival, Condé-Green et al. [57] showed that centrifugation, although very aggressive on adipocytes, cleared the fat of most blood remnants and showed the highest concentration of ASCs in the pellet that could be extracted and added to other substances to increase survival.

The Coleman and microfat techniques could be performed extemporaneously in the operating room by surgeons, allowing their use during the same surgical procedure. Inversely, AD-SVF is extracted, in a few hours, through an enzymatic digestion followed by centrifugation of lipoaspirates, and ASCs isolation requires an in vitro selection and expansion process after plating of AD-SVF (around three weeks), and thus cannot be immediately used in clinical practice. Unlike AD-SVF, ASCs correspond to a homogeneous population of multipotent stem cells. It should also be noted that ASCs can be useful in both allogeneic and autologous treatments, whereas AD-SVF is only suitable for autologous treatments due to the presence of various cell types, such as leukocytes, known to be immunogenic. Moreover, it is important to consider that European regulations classify AD-SVF and ASCs as ATMP (Advanced Therapy Medicinal Products) that must be manufactured in compliance with Good Manufacturing Practice (GMP) in authorized facilities. 

Considering the pleiotropic effects of ASCs, displaying immunomodulatory, angiogenic and antifibrotic capabilities, ASCs-based therapy could counteract the main pathogenic aspects of SSc and might represent a breakthrough in this disease with treatments need. In the context of this chronic autoimmune disease, the use of autologous fat-grafting or cells from autologous adipose tissue requires the proof of the therapeutic potential of these cells to validate autologous approaches. Recently, Magalon et al. [58] showed that the quantitative distribution of the endothelial, stromal, immune, and pericyte cell subsets was preserved within the AD-SVF extracted from SSc patients. Moreover, SSc did not compromise the vascular repair capacity of AD-SVF, endorsing its autologous use. Likewise, Capelli et al. [59] showed that in vitro-expanded ASCs isolated from patients with SSc maintained the same phenotypic pattern, proliferation and differentiation potential. Other studies confirmed that ASCs from SSc patients are phenotypically indistinguishable from similar cells isolated from healthy age- and sex-matched controls and have comparable differentiation capacity. Likewise, our team showed that SSc did not significantly impact the phenotype or differentiation potential [9,60]. In contrast, Di Benedetto et al. reported deregulations of several microRNA, involved in the regulation of senescence and fibrosis development, leading to a profibrotic microRNA profiling of ASCs during SSc [61]. Nevertheless, we showed that ASCs from SSc patients display similar paracrine proangiogenic and antifibrotic capacities of ASCs isolated healthy donors [60]. These heterogenous conclusions are debated in the scientific community [62] and further functional studies are required to fully decipher the impact of the SSc on the therapeutic potential of ASCs.

Thus, these results support the feasibility of treatment approaches based on autologous fat-grafting, AD-SVF, or in vitro-expanded ASCs in a comprehensive way and provide insight into potential mechanisms whereby local effects and disease modification may occur.

## 6. Use of Autologous Fat-Grafting or Adipose-Tissue-Derived Cell Therapies for Facial Handicaps in Systemic Sclerosis

### 6.1. Autologous Fat-Grafting for the Face of Patients Suffering from SSc

Several studies have evaluated the use of autologous fat-grafting to improve both aesthetic and functional facial aspects. All the studies confirm the good safety of autologous fat-grafting for face involvement and encouraging results, especially on the MHISS score [49,52,63,64,65,66]. However, these open-label studies are difficult to compare due to the high heterogeneity in fat grafting procedures and clinical outcome evaluations. Another limitation of these studies is the time frame of the follow-up. Although the effects of fat grafting can be assessed as early as 3 months after injection, there is a need for long-term follow-ups. Almadori et al. [64] also suggested that multiple sequential interventions produce cumulative benefits in both mouth function and aesthetics. In line with this, Pignatti et al. [66] reported positive results in a prospective study evaluating successive adipose tissue transfer injections repeated two or three times every 6 months . Larger and controlled studies are required to better estimate the long-term efficacy of autologous fat-grafting on facial handicaps for patients with SSc. Also, it would be interesting to estimate the residual volume of fat grafting a few months after injection; as Kølle et al. [67] showed a significant improvement in graft survival when adipose tissue was enriched in ASCs.

### 6.2. ASC Injections in Facial Skin-Induced Lesions 

Due to the regenerative properties of ASCs, it has been postulated that direct injection of these cells may offer increased benefits compared to fat grafting. To date, only one study has been carried out. Onesti et al. [68] compared the effects of lipotransfer (*n* = 5) versus the injection of ASCs (*n* = 5) in the mouth functional disability. Unfortunately, from a methodological point of view, numerous limitations can be pointed out, which do not allow for the comparison of results. First, the patients were not randomized. Second, detailed data on the MHISS score are missing, and the severity of skin fibrosis was not assessed. There was no calculation of the number of subjects required to show the superiority or non-inferiority of one technique over the other. Except for maximal mouth opening measurement, the evaluation criteria were subjective, and the follow-up was not blinded. Finally, ASCs were suspended in synthetic stabilized hyaluronic acid solution, whose antiageing effects are also well known. Autologous fat grafting and ASC-based approaches are reported in Table 1 and illustrated in Figure 3.

### 6.3. Combined Platelet-Rich Plasma and Lipofilling in Facial Skin-Induced Lesions 

The clinical success of fat grafting on the face is variable with an unpredictable local survival rate. Thus, pro-survival strategies to improve the maintenance of fat volume and trophicity are sought. Platelet-Rich Plasma (PRP) is a suspension of activated platelets in plasma that release growth factors and is characterized by a higher platelet concentration than in blood. Modarressi [70] and Cervelli et al. [71] documented in in vitro experiments and clinical studies on wounds and soft tissue defects that the use of PRP during fat grafting favors adipose tissue maintenance and survival. 

Virzi et al. [72] confirmed the positive effects of the combined use of autologous fat-grafting and PRP in six SSc patients. The PRP was injected in the perioral and malar areas. After 10 min, in the same PRP injection site, the fat grafting was performed with a 15-gauge cannula. Then, 500 mg of solumedrol was administered to minimize tissue inflammation and postoperative oedema associated with compressive medication. At 3 months, PRP, in combination with autologous fat-grafting, led to a substantial increase in skin elasticity and improved labial rhyme opening rate and vascularization. 

Blezien et al. [73] also reported satisfying results, both functional and aesthetic, in 7 SSc patients. Microfat was left to sediment by gravity, excess oil and blood were eliminated, and then the PRP was added before injection in the lips. At 12 months, a significant increase in the oral opening, improvement in lip thickness, and reduction in the MHISS score were observed without significant, long-term side effects. However, the potential benefits of PRP in SSc need to be better assessed in randomized clinical trials. 

Abellan Lopez et al. [74] reported the safety and efficacy of PRP enrichment to micro-lipofilling protocol treating facial disabilities of SSc in 13 SSc patients. The PRP was added to the micro-fat in a closed system thanks to the Female/Female syringe connector of the St’Rim kit just before injection in order to obtain a final mix of 70% of micro-fat and 30% of PRP. A 22.0% of improvement from baseline of the MHISS score was observed, with a mean decrease of 6.5 points at 6 months. Volumizing and aesthetic effects assessed on standard photographs were also noted. However, they did not record significant changes regarding the facial Rodnan score, mouth opening, and Xerostomia Inventory score at 6 months. 

To summarize, three strategies were evaluated in the treatment of face involvement, fat grafting, ASCs injection and combined PRP and fat grafting. To date, the only results are from observational studies, limited by small sample size and with different outcome measures and short follow-ups. Before and after comparisons show functional and aesthetic improvement without major adverse effects.

## 7. Use of Autologous Fat-Grafting or Adipose-Tissue-Derived Cell Therapy for Hand Handicap in Systemic Sclerosis

### 7.1. Autologous Adipose Tissue Grafting in the Treatment of Hand Manifestations 

To date, the experience of autologous fat-grafting on the hands in SSc disease is limited to small scale open label studies, summarized in Table 2 and illustrated in Figure 4. 

To our knowledge, the study by Bank et al. [54] is the only study to use autologous adipose tissue to improve severe and refractory Raynaud’s phenomenon. The goal of this open-label study was to evaluate the effect of fat grafting on the dorsal and palmar areas of the hands of 13 patients suffering from refractory Raynaud’s phenomenon of which 9 patients had SSc. Treatment showed significant improvement in pain and the number, duration, and severity of cold attacks, with a decrease in DU number. 

Del Papa et al. [52] investigated the therapeutic potential of regional injection of autologous adipose tissue to treat ischemic DUs with a very slow or no tendency to heal. First, in a pilot open study, fifteen SSc patients with only one active ischemic DU, lasting for at least 5 months and showing no tendency to heal were enrolled. The grafting procedure consisted in adipose tissue injection at the base of the affected finger after a small skin incision to slightly reduce the pressure due to the introduction of the adipose tissue. In this pilot study, ulcer healing was observed in all the patients after 2 months. The result was maintained during the following 6-month period and no new DU occurred. Two other monocentric studies on small cohorts of patients confirmed the benefit of fat grafting the hand dysfunction related to SSc [65,66].

To confirm these preliminary results, a monocentric randomized controlled study was conducted [53]. The study was stopped following a preplanned interim analysis after the enrolment of 25 and 13 patients who received autologous fat-grafting and the saline solution, respectively. The results at 8 weeks strongly confirmed that autologous adipose tissue grafting was effective in inducing DU healing as DU healing was observed in 23/25 patients treated with fat grafting, conversely to only 1/13 patients who received the placebo. Interestingly, the 12 patients who received the saline solution underwent a rescue therapy with autologous adipose tissue grafting leading to DU healing for all of them. 

### 7.2. Autologous AD-SVF in the Treatment of Hand Handicap in SSc 

To avoid the risk of ischemia related to the volume effect of adipose tissue with inherent increased pressure during the injection and with the aim of focusing on a trophic effect instead of a volumetric one, our team chose to inject AD-SVF along the fingers of SSc patients in the hope of improving hand disability. 

SCLERADEC [47] was the first study to assess the safety and potential benefit of the administration of autologous AD-SVF in SSc patients’ fingers. Twelve patients with a hand disability (CHFS greater than 20/90) were included in this phase I, monocentric, open-label, single-arm trial. Harvested adipose tissue was directly transferred to a sterile collector bag and immediately transported to the registered Cell Therapy Unit for AD-SVF (5 mL) production using the automated processing Celution 800/CRS system (Cytori Therapeutics). The treatment consisted in injecting 1 mL of AD-SVF diluted by one-half with Ringer’s lactate in each digit, i.e., 0.5 mL on either side of the subcutaneous tissue in contact with the neurovascular pedicles. AD-SVF injection was conducted under conscious sedation. The primary objective was reached as the procedure was feasible for all patients and well tolerated. Exploratory endpoints showed a significant improvement in CHFS, RCS, hand pain, and SHAQ at 6 months. The assessment at 12 months showed long-lasting benefits of AD-SVF injection on all parameters except for hand pain, which was more severe at 12 months, but less than the baseline [75]. The total number of DU decreased at 12 months. The circumference of the fingers and the avascular score using nail fold capillaroscopy were significantly lower compared to the baseline, suggesting an anti-inflammatory and angiogenic effect. Patients observed a significant improvement in their hand strength and mobility. The long-term follow-up data (between 22 and 30 months) for CHFS, SHAQ, RCS, and hand pain endpoints showed significant improvement compared to the baseline, with a decrease in the number of DUs [76]. At the end of the follow-up, only one of the eight patients who had previously received an ilomedine infusion required a new infusion. 

In 2020, Park et al. [77] conducted an open-label phase I clinical trial evaluating the safety and efficacy of AD-SVF administration for the hand disability in 20 patients with SSc. The AD-SVF were processed by a close system (SmartX^®^, DongKoo Bio &Pharma Co., Ltd.) and injected into each finger. Skin fibrosis, hand oedema and quality of life were improved. Furthermore, almost one-third of the active digital ulcers were healed at 24 weeks after injections. 

These encouraging results led to two randomized controlled clinical trials. The STAR trial conducted in the USA was a double-blind, multicentric, phase II randomized controlled study in 88 subjects with significant impairment of hand function (CHFS > 20) [78]. Patients were randomly assigned to AD-SVF treatment (*n* = 48) or placebo (Ringer’s lactate, *n* = 40). The analysis of the primary endpoint (change in CHFS at 24 and 48 weeks after treatment) showed a trend in favour of the AD-SVF group at 24 and 48 weeks without statistical significance. Significant improvements at 48 weeks in the AD-SVF group compared to the baseline were observed for CHFS, SHAQ, and quality of life for patients with dcSSc. Conversely, the results from the double-blind, multicentric, phase II randomized controlled study SCLERADEC II conducted in 40 patients in France showed an improvement of hand function in both groups with no superiority of the AD-SVF administration versus placebo, even in the dcSSc group [79].

To summarize, strategies were evaluated in the treatment of hand involvement, fat grafting and AD-SVF injection. Despite very encouraging preliminary results with AD-SVF, the phase II trials did not demonstrate the superiority of AD-SVF over a placebo. Only Del Papa et al. published the results of a randomized controlled trial showing that fat grafting was significantly more effective in DU healing than the injection of a 0.9% saline solution at the base of the affected finger. In all cases, the autologous fat-grafting procedure and AD-SVF injection performed by experienced surgeons have proven to be safe.

## 8. Discussion

Although the results are extremely promising, this literature review mainly gathers data from observational studies (only three randomized controlled clinical trials) and highlights the great clinical practice heterogeneity. The number of patients treated in each study is relatively low and ranges from 2 to 88, with both patients suffering from lcSSc and dcSS. Besides, clinical follow-up varies from 8 weeks to 53 months. The limited number of studies on the clinical use of autologous fat-grafting and adipose tissue cell-based therapies for face and hands treatment, high heterogeneity due to disparate study designs and measurement of outcomes deemed meta-analytic data reporting inappropriate.

Complications from the donor site are the same as those expected from liposuction—pain and swelling are minimal and there should be very little bruising. Complications of fat grafting are very rare and usually do not develop if the procedure is performed by experienced surgeons. No studies reported major adverse events. If pain was present, it was transient after the surgery (harvest or injection). Infectious complications were rare. Possible harvesting difficulties are related to low BMI. It is often preferable in very thin patients with scleroderma to collect small amounts of fatty tissue in multiple symmetric areas. In addition, general anesthesia may be difficult or even contraindicated due to the development of pulmonary fibrosis in some patients.

It is important to emphasize that only Del Papa's team published the results of a randomized controlled trial showing that adipose tissue grafting was significantly more effective in DU healing than the injection of a 0.9% saline solution at the base of the affected finger [53]. Despite very encouraging preliminary results with AD-SVF, the phase II trials did not demonstrate the superiority of AD-SVF over placebo on the CHFS [78,79]. As many independent factors can impair hand function in SSc patients, the CHFS score, which is a self-completed questionnaire, is not sufficiently accurate to assess the efficacy of AD-SVF administration. In future clinical trials, more targeted and objective primary criteria should be more helpful as DUs. These results also highlight the need for further studies assessing AD-SVF efficacy to include a more homogeneous patient population in terms of disease features (patients with early disease and with diffuse cutaneous form) and hand disability (severity and pattern of hand damage).

Another point raised by this review is the absence of consensus regarding the denominations of adipose-cell-based therapies. However, recommendations from IFATS and ISCT are available, and their applications are also strongly recommended to facilitate the development of international standard-based studies [11]. Del Papa et al. [52] used the term “adipose-tissue-derived cell (ATDCs) fraction” obtained after centrifugation of the lipoaspirate and removal of the upper phase containing the oil supernatant and adipocytes and the lower phase containing blood and plasma residuals. Almadori et al. [64] used the term “stem cell-enriched adipose tissue” to designate the distal portion of the lipoaspirate recovered after centrifugation. However, both terms correspond to the Coleman fat grafting procedure. 

The development of cell-based products like AD-SVF obtained after an enzymatic digestion or cultured ASCs are limited, as they only concern 5 of the 19 studies we reviewed. This is explained by the regulatory constraints of their development according to directive no. 1394/2007 from the European Parliament and the European Council. These therapies are designated as ATMP, meaning they must be produced according to the Good Manufacturing Practices in a cell therapy facility, including the achievement of process validation batches with outstanding characterization of the final product and definition of biological acceptance criteria. Thus, the cost of these therapies will necessarily be significant and should justify an important clinical benefit for SSc patients compared to existing solutions.

Despite the constraints in terms of production, as ASCs have multiple paracrine effects, a multicentric randomized placebo-controlled clinical trial is evaluating whether local administration of allogenic ASCs heal refractory ischaemic DU in SSc patients (ADUSE NCT04356755). This regulatory issue has recently led Tonnard et al. [81] to develop a new lipoaspirate processing technique consisting of mechanical disaggregation by manually transferring the fat repeatedly between two connected syringes followed by a filtration to reduce the size of the fat particles needed to obtain an injectable product, known as nanofat. As the product is not obtained using enzymatic digestion, it does not meet the ATMP definition that should favour its wide dissemination in clinical practice. Mechanical disaggregation causes stromal fragmentation and considerable adipocyte breakage. Therefore, intracellular triglycerides released from broken adipocytes, together with the residual fluid present in the lipoaspirate, formed an emulsion containing stromal cells as aggregates. Thus, the term “emulsified fat” is more accurate, as the true dimensions required for nano-size particles are not reached by this production process. In their pioneer paper, Tonnard et al. [81] found comparable quantities of ASCs in macrofat, microfat and nanofat. Likewise, the ASC cultures of the stem cells from the three samples showed the same proliferation and differentiation capacities. In addition, mechanically disrupted cell aggregates remain attached to their natural matrix niche, which has been shown to promote cell viability, proliferation, and differentiation. Consequently, because of the lack of adipocytes, the volumetric effect of nanofat is obviously limited and nanofat is injected alone at an intradermal level to enhance the quality and elasticity of the skin or could be combined with fat to enhance engraftment. 

The combination of biological products with fat is under investigation and represent 3 of the 19 studies reviewed. These three studies assessed the effects of the combined use of autologous fat-grafting and Platelet-Rich Plasma (PRP). The objective of adding PRP to an autologous fat graft is to increase the survival rate of the graft and to improve the cutaneous trophicity above the grafted areas. Indeed, the exact percentage of facial fat grafts retained is unpredictable and the reported survival rate of adipose tissue varies with different estimation methods. A recent meta-analysis [82] including 27 studies and more than 1000 patients reported a volume retention rate from 26 to 83%, with a mean follow-up of 3–24 months after fat grafting in the face. The authors suggested a trend towards better retention was found for secondary fat grafting procedures. Furthermore, the procedure appeared to be very safe; only 2.8% of all patients had complications. PRP could increase fat graft survival by providing nutrient support from its plasma component and enhancing the proliferation of preadipocytes through the secretion of a great variety of growth factors and cytokines produced by the platelets [83,84]. Due to its ease of use, PRP is a good candidate to potentiate, improve and extend the beneficial effects of fat grafting. However, one of the weaknesses in the use of PRP remains the high heterogeneity from one preparation to another, which could influence clinical outcomes and the potential benefit of SSc must be properly assessed. Recently, Willemsen et al. [85] carried out a randomized, double-blind, placebo-controlled study to assess the benefits of adding PRP to facial lipofilling. Thirteen healthy patients underwent lipofilling with 3 mL of PRP and 12 underwent lipofilling with a placebo (3 mL of sterile saline). The results showed that PRP significantly reduced postoperative recovery time but did not improve patient outcomes with regards to skin elasticity, improvement of the nasolabial fold, or patient satisfaction. A potential bias in this study was the lack of uniform concentrations of platelet-rich plasma added. Indeed, a study suggests that a high PRP concentration converts ASCs into a fibroblast-like phenotype, with increased collagen RNA expression and altered paracrine signalling that negatively influences endothelial vessel formation [86]. Abellan Lopez et al. [74] described the microfat–PRP mixture, showing that a 70/30% mixture corresponded with a facial filler from a rheological point of view [87], whereas an increased dose of platelets within microfat (>500 million platelets/mL of fat) was associated with negative effects on engraftment in a mice study [88]. Both studies only use a pure PRP containing >90% of platelets compared to leukocytes and red blood cells.

ASCs are known to secrete a spectrum of factors (secretome), such as cytokines, chemokines, cell adhesion molecules, lipid mediators, interleukins, growth factors, hormones, exosomes, microvesicles, etc. [89]. ASCs–secretome showed anti-inflammatory, antiapoptotic and immunomodulatory properties [90]. These factors participate in tissue repair and regeneration through their paracrine actions that mediate cell-to-cell signaling. In comparison with MSCs, the secretome is less immunogenic but exerts similar biological actions, so it can be considered an ideal cell-free therapeutic alternative. Extracellular vesicles, including exosomes and microvesicles, derived from MSCs also exert similar effects as their parental cells [91,92]. Velier et al. [60] showed that SSc effects do not significantly compromise the angiogenic and antifibrotic paracrine properties of ASCs. These results encourage clinical evaluation of these emerging strategies and provide additional support for the development of cell-free therapies based on the ASCs’ secretome.

## 9. Conclusions

The popularity of fat grafting and adipose-derived cell therapies has increased over the last decade and offers real hope for SSc patients suffering from facial and hand handicaps—two current disease manifestations that are not sufficiently improved with traditional drugs. Adipose tissue has received a lot of attention due to the ease of harvest and the number of multipotent cells obtainable. Fat grafting remains an easy and accurate procedure when a volumetric effect is required besides the trophic effect. The cells from adipose tissue can be used for therapeutic purposes, considering the advantages or drawbacks of each procedure, from rapidly freshly isolated AD-SVF to culture-expanded ASCs, both requiring the expertise of a dedicated facility. The therapeutic potential of the nanofat deserves to be evaluated in SSc due to the ease of use in the operating room and the cost-effectiveness. 

Cell-free therapy using ASCs’ secretome or extracellular vesicles can also offer an exciting approach in regenerative medicine. To date, the estimated efficacy of fat grafting and adipose-derived cell therapies is mainly based on small cohort studies. Our review shows a lack of robust evidence from controlled trials comparing adipose-derived products with standard care. To facilitate the assessment and comparison of innovative therapies, relevant, objective effectiveness criteria considering the clinical heterogeneity of SSc are needed; for example, DU healing in hand involvement. Furthermore, studies stratified by disease subtypes (dcSSc and lcSSc) and severity might shed light on the effects of these factors. Likewise, in future clinical studies, both clinical (volume and optimal timing for fat grafting) and biological (optimal number and type of cells for adipose-derived products) evaluations are needed to optimize these therapies. In conclusion, randomized controlled trials stratified on subtypes and disease severity remain essential to confirm the therapeutic potential of fat grafting and adipose-derived cell therapies and to study efficacy in relation to the biological characteristics of the product and its long-term effects.

## Figures and Tables

**Figure 1 biomedicines-11-00348-f001:**
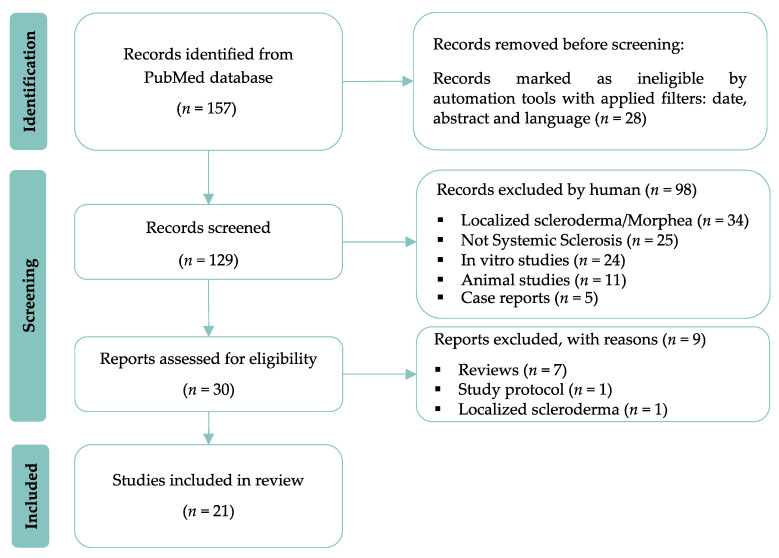
Study attrition flowchart.

**Figure 2 biomedicines-11-00348-f002:**
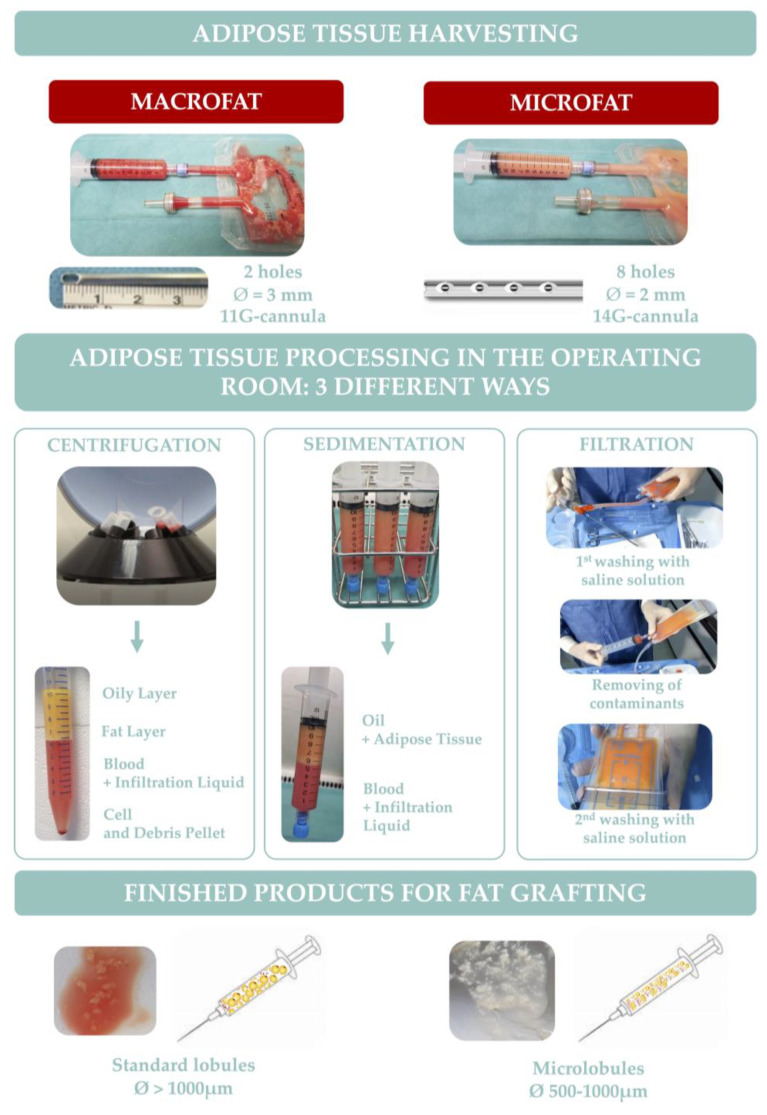
Harvesting, processing and characteristics of macrofat and microfat.

**Figure 3 biomedicines-11-00348-f003:**
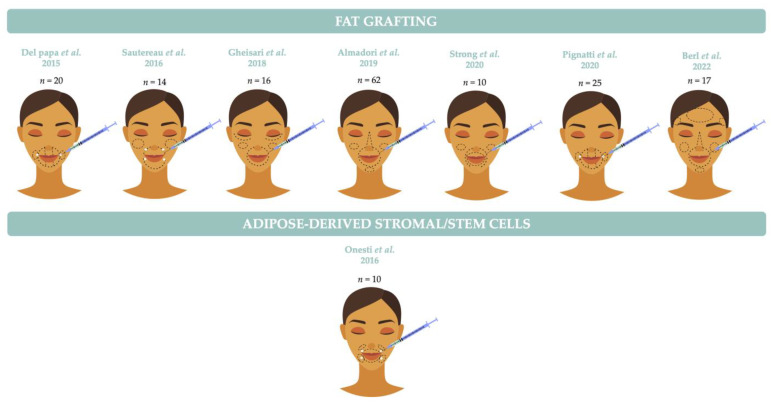
Localization of injections for the treatment of face involvement in systemic sclerosis.

**Figure 4 biomedicines-11-00348-f004:**
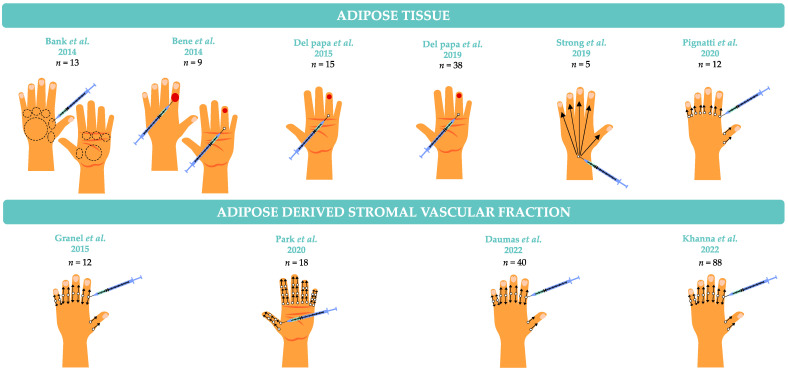
Localization of injections for the treatment of hand involvement in systemic sclerosis.

**Table 1 biomedicines-11-00348-t001:** Studies evaluating autologous fat-grafting and adipose-derived cell therapy products for the treatment of face involvement in SSc.

Publications	Fat Harvesting and Adipose Tissue Processing	Administration	Patient Characteristics	Follow-Up	Clinical Evaluation of Treatment	Side Effects
**ADIPOSE TISSUE**						
**Del Papa** et al. [52] 2015 Open-label study	Coleman’s technique (12G cannula followed by a centrifugation at 700 *g* for 3 min)	Up to atotal amount of 16 mL/patient Injections using a 19G cannula	20 women with dcSSc	1 and 3 months	(1) Increase in **maximum interincisal distance** at 3 months * (2) Gradual increase in **mouth perimeter** measurement from baseline to 3 months * (3) **Durometer measurements in perioral areas**: perioral skin hardness decreased * (4) **Labial capillaroscopy**: increased number of capillaries with a more regular microvascular architecture * (5) **Histopathological findings**: reduction of dermo-epidermal junction flattening, reconstruction of normal ridge pattern and dermal papillae, and increase in microvascular density (6) **Patient satisfaction**: 16 claimed to be very satisfied and 4 satisfied	Small areas of ecchymosis
**Sautereau** et al. [49] 2016 Open-label study	Microfat harvesting technique (14G cannula) Filtration of the microfat in a closed-circuit system Puregraft^®^	Median quantity of fat injected was 17 mL (6.2 to 23 mL) Four entry points: 2 along the nasogenian grooves and 2 along the lower lip Injections using a 21G cannula	14 women 6 lcSSc 8 dcSSc	3 and 6 months	(1) **MHISS score**: mean decrease of 7.3 at 3 months and 10.8 at 6 months * (2) Improvement in **perioral modified Rodnan skin score** * (3) **Mouth opening** increased with a decrease in the VAS assessing discomfort related to mouth opening limitation * (4) **Oral Sicca Syndrome**: Xerostomia Inventory score improved at 6 months. The time to melt a sugar cube on the tongue decreased. VAS focusing on handicaps related to dry mouth decreased at 3 and 6 months * (5) **Facial Pain**: pain induced by palpation of the face muscles and VAS for facial pain decreased * (6) **Cutometer measurements of skin elasticity** showed no significant change (7) **Patient satisfaction**: 9 patients (75%) were very satisfied or satisfied, 2 moderately satisfied, and 1 unsatisfied (8) **Global disability**: no significant change in SSc-HAQ score was observed	Small areas of bruising and local pain Perioral sensitiv emanifestation (*n* = 1) Trigeminal neuralgia (*n* = 1)
**Gheisari** et al. [63] 2018 Open-label study	Liposuction with a 3 mm cannula followed by sedimentation of the adipose tissue for 10 min	15 to 40 mL of fat injected/ patient Injections using an 18G cannula	16 women 6 lcSSc 10 dcSSc	3 months	(1) Increased **Mouth opening** * (2) **MHISS score**: mean decrease of 6.12 * (3) Reduction of **Face Rodnan score** * (4) **Cutaneous resonance running** time showed no significant change (5) **Aesthetic effect**: 13 patients showed an improvement (6) **Patient satisfaction**: 10 patients were very satisfied, 2 somewhat satisfied, and 3 unsatisfied	Bruising at the zone of fat harvest
**Almadori** et al. [64] 2019 Open-label study	Coleman’s technique (3 mm cannula followed by a centrifugation 3000 rpm 3 min)	Mean quantity of fat injected was 10.2 mL Injections using a 14G cannula after small skin incisions	61 women and 1 man 36 lcSSc 26 dcSSc	Mean follow-up after the last treatment 12.41 months (6–53 months) 29 patients received ≤ 2 treatments 33 ≥ 3 treatments	(1) **MHISS score**: mean decrease of 6.87 * with a greater improvement for patients that received ≥3 treatments (2) **Psychological outcomes**: Improvement in all psychological measures: VAS, DAS24 score, HADS-A score, HADS-D score, and BFNE scale * (3) **Volumetric augmentation outcome**: reduction in perioral wrinkling and ridges, improvement in lip volumes and increased vermillion	Bruising, swelling, and tenderness of donor site Superficial wound infection at the recipient site (*n* = 1)
**Strong** et al. [65] 2021 Open-label study	Coleman’s technique (3 mm cannula followed by a centrifugation 3000 rpm 3 min)	Mean quantity of fat injected was 26.1 mL (range 7.5–71 mL) after small incisions along the lateral commissures	10 women	Evaluation at 1, 3, 6 and 12 weeks Mean follow-up = 6.2 months	(1) **Volumetric augmentation** outcome: reduction perioral rhytids, improvement in soft tissue volume (2) **Patient satisfaction**: subjective improvement of skin quality	Minor adverse events (AEs): pain, bruising and swelling at the donor site and minor bruising at the recipient site
**Pignatti** et al. [66] 2020 Open-label study	Coleman’s technique (3 mm cannula followed by a centrifugation at 3000 rpm for 3 min)	A total of 16 mL divided in 8 sites around the mouth into subcutaneous and submucosal planes with retrograde technique	In the perioral area: 10/17 received 2 treatments and 7/17 received 3 treatments at 6 months interval	From 12 to 18 months after the first procedure	(1) **Patient satisfaction**: improvement of perioral skin tension (2) **Oral Sicca Syndrome**: subjective amelioration of xerostomia (3) No significant change of the **mouth opening distance** (4) MHISS score: the trend towards improvement	Small ecchymotic areas around the mouth
**Berl** et al. [69] 2022 Open-label study	Liposuction with a 3 or 4 mm cannula followed by sedimentation of the adipose tissue for 10 min Transformation of macrografts to micrografts by transferring fat between syringues via a stopcock valve	Mean volume injected was 72 mL (range 20–180 mL) Injection using Coleman^TM^ micro infiltration cannulas	17 patients (16 dcSSc,1 lcSSc), 9 had multiple surgeries (MS): 5 had 2MS, 2 had 3MS, 2 had 4MS	1 week, 1 and 3 months	(1) **Mouth opening distance** improved at 3 months (mean of 0.85 cm range 0.2–2.5) (2) **Overall satisfaction** (1–7 scale) was high (mean 5.2) and 88% of patients would be willing to repeat the procedure	Postoperative hematoma (*n* = 3) and postoperative pain (*n* = 10)
**ADIPOSE-DERIVED STROMAL/STEM CELLS**						
**Onesti** et al. [68] 2016 Open-label study	Liposuction with a one-hole 3 mm cannula followed by sedimentation of the adipose tissue for 15 min	Two injections of treatment at 3-month intervals **ADSCs group**: injection in perioral area of 32 × 10^5^ cells in 4 mL of hyaluronic acid with a 30G ½ needle **Fat grafting group**: injection of 16 mL in the perioral region at sub-cutaneous level with a 2 mm cannula	10 patients ASCs group: 5 patients Fat grafting group: 5 patients	1 week, 1 month and 1 year	(1) Increase in **maximum interincisal distance** but no significant difference between groups (2) **MHISS score**: improvement but no statistically significant difference between groups (3) **VAS scale**: improvement but no significant difference between groups (4) **Patient satisfaction**: in the ASCs group, 20% of the patients were satisfied, 80% very satisfied vs. in the fat grafting group, 80% of patients were satisfied and 20% very satisfied	No information about safety

* Significant results (*p* < 0.05).

**Table 2 biomedicines-11-00348-t002:** Studies evaluating autologous fat-grafting and adipose-derived cell therapy products for the treatment of hand involvement in SSc.

Publications	Fat Harvesting and Adipose Tissue Processing	Administration	Patient Characteristics	Follow-Up	Clinical Evaluation of Treatment	Side Effects
**ADIPOSE TISSUE**						
**Bank et al.** [54] 2014 Open-label study	Coleman’s technique and sedimentation	Median quantity of fat injected was 26 mL (11 to 30 mL) Injections using an 18G cannula after an incision made with an 18G needle	13 patients with Raynaud phenomenon including 9 patients with SSc	Average follow-up of 17.9 months	(1) **Raynaud phenomenon pain**: improvement in pain * (2) **Cold attacks**: improvement in number, duration and severity * (3) Twelve hands (57%) had at least one **DU** before treatment compared to only 5 (24%) after treatment * (4) Increased **blood flow** per imaging noted in 5 of 11 hands tested	Recurrence of ulceration with increased severity of Raynaud’s attacks (*n* = 1), Cellulitis (*n* = 1), Transient digital numbness (*n* = 1)
**Bene** et al. [80] 2014 Open-label study	Coleman’s technique (3 mm cannula followed by a centrifugation 3000 at rpm for 3 min)	2–3 mL of fat injected/affected fingers, at the border of the larger ulcers with different depths, or at the finger base for the smaller digital ulcers, with blunt cannula	9 patients for a total of 15 digital ulcers, refractory to pharmacological treatment and wound care. 5 lcSSc 4 dcSSc	From 6 to 24 months Clinical outcomes were evaluated at 3 months	(1) **DU Healing**: 10/15 DU, 5/9 patients (2) **Ulcer reduction >50%:** 2 (3) **Pain reduction**: 7/9 patients	No ulcer recurrence in the same location No adverse reactions at the sites of injection
**Del Papa** et al. [52] 2015 Open-label study	Coleman’s technique (18G cannula followed by centrifugation at 920 *g* for 3 min)	0.5 to 1 mL of fat injected at the base of the affected finger under local anesthesia, using an 18G cannula A small skin incision was made at the site of injection	15 women with only one active DU more than 5 months old prior to enrollment 8 lcSSc 7 dcSSc	1, 3, and 6 months	(1) **Time to healing of cardinal ulcers**: mean time was 4.23 weeks (range 2–7), no new DU appeared in any of the treated patients (2) **DU pain severity (VAS):** reduction in pain intensity * The amounts of analgesic drugs drastically decreased after 1 week and none of the patients were still taking analgesics after 1 month (3) **Nailfold Video Capillaroscopy**: an increase in the capillary count was observed *	No
**Del Papa** et al. [53] 2019 Monocentric randomized controlled study	Coleman’s technique (18G cannula followed by a centrifugation of 920 *g* for 3 min) Sham procedure consisting in false liposuction followed by the injection of saline solution at the base of the affected finger.	0.5 to 1 mL of **adipose tissue** injected at the base of the finger with the DU, using an 18G cannula **Sham procedure** consisting of injection of saline solution	38 patients with only one active ischemic DU lasting for at least 6 weeks 25 patients (23 women, 16 dcSSc) treated with autologous adipose tissue and 13 women (7 dcSSc) received saline solution	4 and 8 weeks	(1) **Prevalence of patients in whom DU healing was observed within 8 weeks**: 92.0% of patients actively treated achieved DU healing vs. only 1 of the 13 patients who received saline solution * (2) **DU pain severity (VAS):** reduction in pain intensity in actively treated patients * (3) **Nailfold Video Capillaroscopy**: an increase in capillary numbers in the affected finger was observed in the actively treated patients *	No
**Strong** et al. [65] 2021 Monocentric open-label study	Coleman’s technique (3 mm cannula followed by a centrifugation at 3000 rpm for 3 min)	Mean volume of 53.2 (range 30–78 mL) of fat injected into the bilateral hands	10 women 5 patients received fat grafting to the hands	Evaluation at 1, 3, 6 and 12 weeks Mean follow-up = 6.2 months	**(1) Skin elasticity**: significant subjective improvement **(2) Hand volume**: significant subjective improvement (3) Some improvement in **mobility**	Minor AEs: pain, bruising and swelling at the donor site and minor bruising at the recipient site
**Pignatti** et al. [66] 2020 Monocentric open-label study	Coleman’s technique (12G cannula followed by a centrifugation at 3000 rpm for 3 min) Procedure repeated every 6 months for 2 or 3 times.	0.5 to 1 mL at the base on each side of each finger, for a total up to 10 mL/hand after skin access created by 19G needle	12/25 patients received 3 injections in the fingers at 6-month intervals	From 12 to 18 months after the first procedure	(1) **Complete healing of DU** in 8/9 patients (2) Significant improvement of **hand tension** (3) Differences in **clinometric measures** not significant (4) **RCS**: trend towards improvement (5) **mRSS**: not significantly improved (6) No significant decrease in **pain** (SF-MPQ) and affective descriptors	Temporary oedema and paresthesia in the hands of 2 patients
**ADIPOSE DERIVED–STROMAL VASCULAR FRACTION**						
**Granel** et al. [47] 2015 Monocentric open-label study	Coleman’s technique (3 mm cannula) AD-SVF processed with Celution 800/CRS system (Cytori Therapeutics)	Injection with a 25G cannula of 0.5 mL of AD-SVF into each lateral side of each digit, both hands.	12 patients with a hand disability (CHFS > 20/90)	6 months	(1) **CHFS**: 47.4% and 56.0% decrease at M2 and M6 in comparison to baseline was observed (*p* < 0.001) (2) **RCS**: improvement * (3) **Hand pain (VAS)**: improvement * (4) **SHAQ**: improvement * (5) **Numbers of DU**: decrease in total number of DU (6) **Circumference of the fingers, MRSS focused on hands et global MRSS** decreased from baseline * (7) **Capillaroscopy evaluation**: changes observed (8) **Hand strength and mobility**: improvement	4 minor AEs potentially related to the procedure: abdominal bruises (*n* = 2), transient paresthesia (*n* = 1), and located pain (*n* = 1)
**Park** et al. [77] 2020 Monocentric open-label study	Suction-assisted Coleman technique AD-SVF processed in a closed system SmartX^®^ kit (DongKoo Bio & Pharma Co., Ltd.)	Injection of 0.6 mL/finger, at the proximal in 3 points from the metacarpophalangeal joint to the tip of the finger. Digital nerve block method was used	18 patients 8 dcSSc 10 lcSSc	2, 6, 12 and 24 weeks	(1) **mRSS**: significant improvement during the follow-up * (2) **mRSS** applied to hands: significant improvement at 6, 12 and 24 weeks * (3) **RCS**: slight improvement (4) **CHFS**: increase tendency (5) **Hand oedema**: significant decrease in mean circumference of both hands at 24 weeks * (6) **DU**: 31.6% were healed at 24 weeks (7) **Disease-related QOL**: improvement (8) Kapandji score, Hand pain (VAS), capillaroscopy evaluation: no change	5 minor AEs: transient paresthesia (*n* = 1), dropout due to dizziness after lidocaine injection (*n* = 1), transient pallor in fingers (*n* = 3),
**Daumas** et al. [79] 2022 Double-blind, multicentric, phase II randomized controlled study	Coleman’s technique (3 mm cannula) AD-SVF processed with Celution 800/CRS system (Cytori Therapeutics)	Injection with a 25G cannula of 0.5 mL of AD-SVF into each lateral side of each digit, both hands.	40 patients **AD-SVF group**: 20 (10 dcSSc and 10 lcSSc) **Placebo group**: 20 were (5 dcSSc and 15 lcSSc).	7 days, 1, 3 and 6 months	(1) Improvement of **CHFS score** at 3 and 6 months in both groups but no difference between groups (2) Improvement of the **SHAQ, hand pain severity (VAS), mRSS applied to hands** and **Raynaud’s phenomenon** severity at 3 and 6 months without significance between groups (3) **Mean number of healed ulcers/patients** was lower for the AD-SVF versus placebo groups	Severe AEs related to the worsening of the underlying SSc (AD-SVF group: *n* = 7) Severe AE attributed to the surgical procedure (placebo group: *n* = 1)
**Khanna** et al. [78] 2022 Double-blind, multicentric, phase II randomized controlled study	Liposuction using manual aspiration (no information about the cannula type) AD-SVF processed with Celution 800/CRS system (Cytori Therapeutics)	**Placebo**: Ringer’s lactate + 0.1–0.2 mL of the patient’s blood **Treatment**: 0.5 mL containing 4 million viable nucleated cells/finger into each lateral side of each digit, both hands using a 25G needle.	88 patients AD-SVF group: 48 patients (32 dcSSc and 16 lcSSc) Placebo group: 40 patients (19 dcSSc and 21 lcSSc)	48 weeks	(1) **CHFS**: numerically higher for the AD-SVF group vs placebo group (2) **HAQ**: greater improvement for dcSSc group (3) **RCS**: greater improvement for dcSSc group	Severe AE in the AD-SVF group: pneumonia (*n* = 2) AE in the placebo group: anemia, hypotension, joint effusion, angina, upper GI tract hemorrhage (*n* = 8)

* Significant results (*p* < 0.05), ** significant results (*p* < 0.001).

## Data Availability

Not applicable.

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
