# Peer review of "Adipose Tissue and Adipose-Tissue-Derived Cell Therapies for the Treatment of the Face and Hands of Patients Suffering from Systemic Sclerosis"

_biomedicines, 2023, doi:10.3390/biomedicines11020348_

Round 1
Reviewer 1 Report
I consider the article sent for review extremely valuable—a fascinating topic requiring much knowledge from the authors.
Systemic sclerosis (SSc) is an autoimmune disease affecting connective tissue cells. It is a rare, chronic disease with a non-uniform course. It is characterized by progressive fibrosis of the skin, blood vessels, and internal organs: kidneys, lungs, heart, and gastrointestinal tract. The blood supply to the tissues of the organs decreases, which damages their structure and impairs their functioning.
Abstract. The goal presented here sounds strange: "The goal of this review is to update the literature on the use of these biological products for face and hands treatment in the context of SSc."... It has to be changed. Besides, I propose to reformulate some sentences here. I especially recommend focusing on these 4 points, some of which sound strange: "the need to use a standardized core set of response measures taking into account the complexity and..." What do the authors mean here?
In Introduction
However, a slightly larger description of SSc would be helpful to clarify what lies at the root of this disease; then, it will be easier for the reader to understand what's going on with this adipose tissue and adipose tissue-derived cell therapies ... The current description is too short. What are the current criteria for the diagnosis of SSc?
I consider the selection of articles correct and the study flowchart convincing that all critical articles were included in this review article.
My final point concerns conclusions. They are a bit "fuzzy." I suggest rewriting them. Besides, the authors should precisely indicate the direction for further research.
Reviewer 2 Report
This is an interesting and important review on the adipose tissue-derived cell therapies for systemic sclerosis.
This review is well organized. Several points below should be addressed and commented.
1. How long does the grafted or injected adipose tissue maintain its function? Re-operation is needed?
Since SSC is a progressing disease, sclerosis might extend into the grafted (injected) adipose tissues?
2.Does fibrosis occur in ASC or AD-SVF grafted or injected? That should be commented.
3.Is there any possibility that the SSC patients have lost the sites for getting AD-SVF?
Reviewer 3 Report
In this review, Zavarro et al summarized adipose tissue and adipose tissue-derived cell therapies for face and hands treatment of patients suffering from systemic sclerosis. In particular, they highlight 1) the need to use a standardized core set of response measures taking into account the complexity and clinical heterogeneity of SSc in order to facilitate assessment and comparison of innovative therapies, 2) it reveals some impacts of the disease on fat grafting success, 3) an important heterogeneity was noticed regarding manufacturing of the adipose-derived products, 4) it shows a lack of robust evidence from controlled trials comparing adipose-derived products with standard care. This review is very useful for readers as it summarizes a great deal of literature in an easy-to-understand manner. I have some questions.
major revisions)
1) You have summarized a lot of literature by systematic review. It would be good to have some additional meta-analysis to compare each study. Would it be possible to examine the usefulness of the treatments using some common metrics?
minor revisions)
1) In line 180, where it says "0 no mouth disability" does it mean "0 (no mouth disability)"?
2) In line 258, I think "strafey" means "strategy".
Reviewer 4 Report
The work presents an up-to-date review of publications related to adipose tissue and adipose tissue-derived cell therapies in the treatment of for face and hands involvement in systemic sclerosis.
The manuscript is very thorough and presents in detail the methodology of publication review, acquisition techniques od adipose tissue, tools for assessing skin involvement and hand function in theardzin. Tables and figures are a valuable supplement.
There are minor issues for considerstion:
- in the paragraphs concerning the effects of treating individual symptoms with the use of the analyzed methods of therapy, a short summary of the quoted results would be useful at the end of each paragraph
- additionally, the discussion should refer to the safety of therapy and possible difficulties in collecting the material, e.g. skin fibrosis, impaired healing.
- Were the analyzed methods also used in limited scleroderma (morphea), in particular in the linear form en coup de sabre?
